Evidence for continual hybridization rather than hybrid speciation between Ligularia duciformis and L. paradoxa (Asteraceae)

Zhang Rong 1 2 3 4
Gong Xun gongxun@mail.kib.ac.cn 1 2
Folk Ryan 5
1 Key Laboratory of Plant Diversity and Biogeography of East Asia, Kunming Institute of Botany, Chinese Academy of Sciences , Kunming , China
2 Department of Economic Plants and Biotechnology, Yunnan Key Laboratory for Wild Plant Resources, Kunming Institute of Botany, Chinese Academy of Sciences , Kunming , China
3 Germplasm Bank of Wild Species, Kunming Institute of Botany, Chinese Academy of Sciences , Kunming , China
4 University of Chinese Academy of Sciences , Beijing , China
5 Florida Museum of Natural History, University of Florida , Gainesville , FL , USA
Riesgo-Escovar Juan
Electronic publication date: 2017 Oct 11
Publication date: 2017
Volume: 5
Electronic Location ID: e3884
Received 2016 Dec 12; Accepted 2017 Sep 13
Copyright: ©2017 Zhang et al.
Copyright year: 2017
Copyright holder: Zhang et al.
License: This is an open access article distributed under the terms of the Creative Commons Attribution License, which permits unrestricted use, distribution, reproduction and adaptation in any medium and for any purpose provided that it is properly attributed. For attribution, the original author(s), title, publication source (PeerJ) and either DOI or URL of the article must be cited.
License URL: https://creativecommons.org/licenses/by/4.0/

Keywords: Ligularia, cpDNA, nrITS, Natural hybridization, SSR and ISSR loci

Funding: National Natural Science Foundation of China 31470336 31600178 This research was supported by the National Natural Science Foundation of China (Grant No. 31470336 to XG and 31600178 to J-J.Y.). The funders had no role in study design, data collection and analysis, decision to publish, or preparation of the manuscript.

==============================
Background

Hybrids possess phenotypic traits that are often intermediate between their parental taxa, which commonly serves as evidence of hybridization in morphological analyses. Natural hybridization has been shown to occur frequently in Ligularia (Asteraceae). In a previous study, Ligularia ×maoniushanensis was demonstrated as a natural hybrid species between L. duciformis and L. paradoxa based on morphological and reproductive traits.

Methods

We used three chloroplast (cpDNA) fragments (psbA-trnH, trnL-rpl32 and trnQ-5′rps16), the nuclear ribosomal internal transcribed spacer (nrITS), and co-dominant SSR and dominant ISSR markers to study natural hybridization between L. duciformis and L. paradoxa growing sympatrically in two locations. Parental taxa were inferred using network analyses of cpDNA and nrITS haplotypes. Admixture among individuals was examined using the Bayesian clustering programs STRUCTURE and NewHybrids based on the SSR and ISSR data; and potential introgression in the SSR loci was assessed using the INTROGRESS package.

Results

The putative parental species were clearly distinguished from other sympatric Ligularia species by nrITS data, and L. ×maoniushanensis individuals were confirmed to be the hybrid offspring of L. duciformis and L. paradoxa. Moreover, introgression was detected among several individuals morphologically identified as L. duciformis or L. paradoxa. Analyses of the cpDNA data revealed primarily unidirectional hybridization between L. duciformis and L. paradoxa, with L. paradoxa as the maternal parent in Mt. Maoniu, whereas bidirectional but asymmetrical hybridization was inferred to occur in Heihai Lake. The STRUCTURE analyses based on the SSR data detected two distinct clusters among the three taxa. The NewHybrids analyses showed that individuals circumscribed as L. ×maoniushanensis were dominated by early- and later-generation and backcrossing hybrids. The NewHybrids results based on the ISSR data were congruent with SSR results. In addition, introgression was detected in some SSR loci, and heterogeneity among loci was found in terms of detected patterns of introgression.

Conclusions

Our data provide strong evidence for hybridization and introgression between L. duciformis and L. paradoxa. Ligularia ×maoniushanensis was demonstrated to be of hybrid origin. Since no evident reproductive isolation was found between the two parental species, detected hybrids appear to be part of hybrid swarms resulting from frequent and ongoing gene flow, which might impede the formation of a new hybrid species.

Introduction

Natural hybridization, which increasingly appears to play a key role in speciation, has been frequently reported between closely related species exhibiting sympatric or parapatric distributions, especially in plants (Abbott et al., 2010; Abbott et al., 2013; Arnold, 1997; Hegarty & Hiscock, 2005; Soltis & Soltis, 2009). Interspecific hybridization may lead to speciation by breaking up and recombining parental genomes (Arnold, 1992; Barton, 2001; Gong, 2005). Resultant hybrids which have been adapted to habitats in the hybrid zone may further recombine parental traits to produce novel characters. They may gradually become isolated reproductively, genetically and ecologically from their parents, eventually forming a hybrid species (Ungerer et al., 1998). For instance, Pinus densata is a highly successful hybrid between P. tabuliformis and P. yunnanensis (Wang & Szmidt, 1994; Wang, Szmidt & Savolainen, 2001; Song et al., 2002; Mao, Li & Wang, 2009; Mao & Wang, 2011; Xing et al., 2014). However, hybrid speciation often is thought not to take place instantaneously; instead, it may be a long-term, gradualistic population process. In such a process, initial generations of hybrids frequently backcross to their parents, causing them to fail to establish themselves as independent lineages with stable trait combinations; this situation is termed introgressive hybridization, and such mixed taxa may form a hybrid swarm (Nolte & Tautz, 2010), in which event these hybrids generally would not be treated as new species (Zhou, Shi & Wu, 2005).

The genus Ligularia (tribe Senecioneae, Asteraceae) consists of approximately 140 species, with 89 species endemic to China (Liu & Illarionova, 2011; Liu, Deng & Liu, 1994), especially in the eastern Qinghai-Tibet Plateau (QTP) region (Jeffrey & Chen, 1984). The QTP region is considered the centre of species diversification and modern-day distribution in Ligularia (Liu, Deng & Liu, 1994). The geographical and ecological heterogeneity of this region could lead to habitat fragmentation, isolation, and rapid continuous hybridization among Ligularia species, which may give rise to new species in the region (Liu, 2004; Liu et al., 2006). Ligularia in the QTP has been used previously to study introgression and interspecific hybridization, and sympatric Ligularia species show close affinity (Pan et al., 2008; Yu, Kuroda & Gong, 2011; Yu et al., 2014; Yu, Kuroda & Gong, 2014). Moreover, the infrageneric phylogeny of this genus also indicates a lack of monophyly, suggesting a need to reassess species relationships (Liu et al., 2006; Pelser et al., 2007; He & Pan, 2015).

Ligularia duciformis and L. paradoxa belong to the series Retusae, which widely distributing throughout the Hengduan Mountains in Yunnan, China (Liu, 1989). According to the field investigations, L. duciformis and L. paradoxa are primarily distributed sympatrically in two areas: Mount Maoniu (hereafter Mt. Maoniu) and Heihai Lake. At Mt. Maoniu, they grow in the damp soil under moist forest with an elevation range 4,000 to 4,100, while at Heihai Lake, they grow in the arid rock crevices in riparian situations, with an elevation range from 4,100 to 4,200 m. The main morphological differences between L. duciformis and L. paradoxa are in leaf shape and floret structure; these two species are accompanied by some morphologically intermediate individuals in the two areas. In the previous study of Pan et al. (2008), these morphologically intermediate individuals in Mt. Maoniu were described as L. ×maoniushanensis, a hybrid species between L. duciformis and L. paradoxa, using morphological, reproductive traits, and molecular markers (ISSR). Among the two parents, L. paradoxa was determined as the female parent based on phylogenetic analyses using the chloroplast trnL-trnF region. In addition, there is another Ligularia species, L. lamarum, belonging to the series Ligularia, sympatrically distributed with L. duciformis and L. paradoxa in both areas.

In this study, we sequenced the nrITS region, three chloroplast (cpDNA) intergenic spacers (psbA-trnH, trnL-rpl32 and trnQ-5′rps16) sampled from two hybrid swarms. Moreover, 11 nuclear SSR markers and 9 ISSR markers were used to reveal the genetic structure of two hybrid swarms, and potential introgression in the SSR loci. This study aimed to address the following questions: (1) Are morphologically intermediate individuals the hybrid progeny of L. duciformis and L. paradoxa? Does the sympatric species L. lamarum participate in hybridization; (2) If hybridization occurs, what is the directionality of this process among parental taxa; (3) Is there any evidence for introgressive hybridization, and if so what is the genetic structure of the hybrid swarms; (4) Is L. ×maoniushanensis a cohesive hybrid species, or do the individuals currently described as L. ×maoniushanensis instead constitute a recurrently formed hybrid swarm?

Materials and Methods

A total of 107 individuals were collected from two hybrid swarms (Mt. Maoniu and Heihai Lake, Ninglang County, Yunnan, China) during August 2013 and June 2014. Among them, 35 and 40 individuals were identified as L. duciformis and L. paradoxa, respectively, according to Flora of China (Liu & Illarionova, 2011), and 19 morphologically intermediate individuals were identified as putative hybrids. In addition, we also sampled 13 sympatric L. lamarum from these two locations to confirm whether it participated in hybridization. All specimens included in this study are summarized in Table S1, and the vouchers were deposited in the Herbarium of Kunming Institute of Botany, Chinese Academy of Sciences (KUN). Putative hybrids were distinguished from parental taxa based on leaf shape and floret morphology (Table 1, Fig. 1). The degree of palmatisect lobation of the leaf blades in putative hybrids differed, ranging from moderately lobed to deeply lobed. Ligularia lamarum was distinguished from L. duciformis and L. paradoxa in size and shape of leaf blades as well as the colour and length of the pappus. Young and healthy leaves collected in the field were dried in silica gel immediately for later DNA extraction. Total genomic DNA was extracted using the modified CTAB (cetyltrimethyl ammonium bromide) method (Doyle, 1991).

Table 1 Key morphological comparison of L. paradoxa, L. duciformis, L. ×maoniushanensis and L. lamarum.

Taxon	Inflorescence	Leaf blades	Pappus	
Ligularia paradoxa	Corymbs	Orbicular or broadly ovate, 3 to 8 palmatisect	Brown, shorter than tubular corolla and longer than its tube, usually deciduous	
L. ducifomis	Compound corymbs	Reniform or cordate, margin irregularly dentate	White or lower part yellow, as long as tube of tubular corolla	
L. ×maoniushanensis	Corymbs	Broadly ovate, palmately lobed to middle	Wine-colored, white at base, shorter than tubular corolla and longer than its tube	
L. lamarum	Racemose	Triangular-sagittate or ovate-cordate	Yellowish, slightly shorter than tubular corolla	

Figure 1 Putative hybrids and parental taxa in nature.

A, B, C and D, respectively, are L. ×maoniushanensis, L. duciformis, L. paradoxa, and L. lamarum.

Sequencing of the three chloroplast intergenic spacers and one nrITS region

Three chloroplast DNA regions were amplified for all the sampled individuals using universal primer pairs for trnL-rpl32 (Shaw et al., 2007), trnQ-5′rps16 (Shaw et al., 2007) and psbA-trnH (Sang, Crawford & Stuessy, 1997; Tate & Simpson, 2003). Polymerase chain reaction (PCR) was performed using a reaction volume of 20 µL, containing 20–60 ng of template DNA, 2.0 µL of 10× PCR buffer (Mg2+ free), 1.0 µL of MgCl2 (25 mM), 1.0 µL of dNTP (10 µM each), 1.0 µL of BSA (20 mg/mL), 0.3 µL of each primer and 0.15 µL of 1.5 units of Taq polymerase (Takara, Dalian, China). Amplification proceeded as follows: an initial 5 min at 95 °C for denaturation; 30 cycles of 45 s at 94 °C; 45 s at 53 °C; and 50 s at 65 °C; followed by a final extension of 7 min at 65 °C. The nrITS sequences of all sampled individuals (107) were amplified using the primers ITS4 and ITS5 (White et al., 1990) using the PCR procedure of Yu, Kuroda & Gong (2014). PCR products were purified by electrophoresis in a 1.2% agarose gel, from which the products were recovered with an E.Z.N.A. Gel Extraction Kit (Omega, Guangzhou, China). All accessions were subjected to sequencing with the amplification primers in an ABI 3700 DNA sequencer with a BigDye Terminator Cycle Sequencing Kit (Applied Biosystems, Foster City, CA, USA).

For the nrITS sequences, direct sequencing was successful for 51.42% of L. duciformis individuals (18/35), 85% of L. paradoxa (34/40) individuals and 100% of L. lamarum individuals. For individuals with chimeric or unreadable peaks in the chromatograms, including each parent-specific and putative hybrid-specific ITS sequence, positive clones with accurate inserts were confirmed using colony PCR (Yu, Kuroda & Gong, 2014). Two to eleven clones were sequenced using ITS4 and ITS5 primers. Both strands of the ITS clones were sequenced. The number of cloned sequences for each accession was listed in Table S2. The DNA sequences generated in this study are available in GenBank with accession numbers: KY307306–KY307785.

Network analyses based on nrITS and cpDNA regions

All sequences were assembled and aligned using SeqMan (DNAstar 7.1, DNASTAR Inc., Madison, WI, USA) and adjusted manually with BioEdit 7.0.4.1 (Hall, 1999). Variable sites and haplotypes were obtained using the DnaSP 5.0 program (Rozas et al., 2003). Haplotype networks for the nrITS sequence and chloroplast DNA fragments were constructed using TCS 1.21 (Clement, Posada & Crandall, 2000) with parsimony probability set to 98%.

SSR genotyping

We selected 11 nuclear SSR markers from previous studies (Ahrens & James, 2013; Liu et al., 2004; Mao, Li & Wang, 2009) that could be successfully applied to sampled individuals in this study, among which loci with variable flanking regions were excluded to avoid null alleles. Moreover, individuals with more than two missing loci were removed from further analyses and a total of 89 individuals (L. duciformis, L. paradoxa and L. ×maoniushanensis) from two locations were used for SSR genotyping. PCR was optimally performed in a reaction volume of 25 µL containing 20–60 ng of DNA, 2.5 µL of 10× PCR buffer, 2 µL of MgCl2 (25 mM), 1 µL of dNTPs (10 mM), 0.5 µL of each primer, 1.0 µL of BSA (20 mg/mL), and 0.15 µL of Taq DNA polymerase (5 unit/µL; Takara, Shiga, Japan). PCR was performed in a thermocycler under the following conditions: 1 cycle at 94 °C for 5 min, which was followed by 35 cycles at 94 °C for 60 s, 53–57 °C for 60 s and 72 °C for 90 s, followed by 1 cycle at 72 °C for 8 min. The PCR products were electrophoretically detected on a 2% agarose gel. We then performed preliminary screening for the SSR loci. An individual was considered null (no amplification) and treated as missing data if amplification failed more than two times at a locus. Individuals containing three or more missing data were excluded from the subsequent analyses.

SSR data analyses

Raw dataset editing and formatting were executed in GenAlEx 6.3 (Peakall & Smouse, 2006). A Bayesian approach was implemented in STRUCTURE 2.3.4 (Pritchard, Stephens & Donnelly, 2000) to identify population structure and potentially admixed individuals from multi-locus data. This Bayesian clustering algorithm identifies an optimal number K of genetic clusters of sampled individuals and simultaneously assigned the individuals to the genetic clusters by calculating the posterior probability of cluster membership. To determine the optimal number of clusters for putative parents and hybrids, 10 independent runs for each K value (K = 1–6) were executed with 106 Markov Chain Monte Carlo iterations and a burn-in period of 100,000. The optimal K value was estimated using the mean value (ΔK) method (Evanno, Regnaut & Goudet, 2005). We adopted a threshold value of membership coefficient (qi) to identify purebred parental individuals (0 < qi < 0.10 or 0.90 < qi < 1.0) and hybrids (0.10 < qi < 0.90; Hoban et al., 2009; Vähä & Primmer, 2006). The software NewHybrids 1.1 (Anderson & Thompson, 2002) was used to assign sampled individuals to six genotype categories (pure species A, pure species B, F1 hybrid, F2 hybrid, F1 backcross to pure species A, and F1 backcross to pure species B) based on the 11 loci. The posterior probability (PP) of categorical membership for each individual was computed using a Bayesian approach. We use PP ≥ 0.90 to designate purebreds. For hybrids, PP ≥ 0.50 to assign a genotypic class; if PP ≤ 0.50, the individual was not assigned to any category.

In STRUCTURE and NewHybrids analyses, given that we wanted to ask whether the three taxa of Ligularia are part of one large panmictic species complex, we jointly analysed the SSR data of individuals from the two sampling sites to effectively exclude substantial genetic differentiation among the populations from different regions; to assess sensitivity of analyses to this procedure, samples from the two sampling sites were also analysed separately.

Introgressive patterns and genomic cline analyses of SSR loci were performed using the INTROGRESS package (Gompert & Buerkle, 2010) in R 3.1.2 (R Core Team, 2012). This package identifies each locus that deviates from expectations of neutral introgression using the log likelihood ratio given the observed data, from which a P value for the significance test is calculated. Significant deviations from neutral expectations for genomic clines were adjusted for multiple comparisons using the false discovery rate (FDR; Benjamini & Hochberg, 1995). We then used the clines.plot function to plot the genomic cline of each locus. Additionally, we estimated the hybrid index for the admixed individuals using functions est.h and mk.image to provide a clear visualization of the variation in introgression and ancestry across loci and for each of the admixed individuals (Gompert & Buerkle, 2010). Ligularia duciformis, L. paradoxa and L. ×maoniushanensis are recognized as homozygotes (Ad/Ad, Ap/Ap) or heterozygote (Ad/Ap) genotypes, respectively.

ISSR screen and analyses

We screened nine polymorphic ISSR marker primers (University of British Columbia: UBC807, UBC808, UBC811, UBC818, UBC828, UBC845, UBC849, UBC850, and UBC857) from a set of 100 ISSR for the sampled individuals of the putative parental species and hybrids. These markers were selected for further amplification. Final PCR amplifications were performed in a reaction volume of 20 µL, containing 30–50 ng of template DNA, 2.0 µL of 10× buffer, 1.6 µL of MgCl2 (25 mmol/L), 1.2 µL of dNTPs (10 mmol/L), 0.3 µL of each primer (10 µmol/L), and 0.15 µL of Taq DNA polymerase (5 U/µL; Takara, Shiga, Japan). The PCR amplification conditions were as follows: 1 cycle at 94 °C for 7 min; 35 cycles at 94 °C for 45 s, 52/53 °C for 1 min and 72 °C for 2 min; followed by 1 cycle at 72 °C for 7 min. Because the possibility of L. lamarum participating in hybridization was excluded according to nrITS and cpDNA data (see Results), only 89 individuals (including 35 L. duciformis, 19 L. ×maoniushanensis, and 35 L. paradoxa individuals) were amplified in each PCR run. For each maker, if the PCR failed for more than five individuals, the marker was excluded from the subsequent analyses. The ISSR bands observed after electrophoresis on an agarose gel were manually recorded as binary characters (0 for absence, 1 for presence and “-” for failed amplification). Only polymorphic band data were used for downstream analysis, since monomorphic bands cannot discern relationships among individuals. The analyses of ISSR data were performed in NewHybrids 1.1 following Yu, Kuroda & Gong (2011).

Results

nrITS sequence analyses

Among 107 sampled individuals, 66 samples were directly sequenced and the remaining 41 samples (including 19 L. ×maoniushanensis, 16 L. duciformis and 6 L. paradoxa) were sequenced by cloning (Table S2). The aligned nrITS sequences had a length of 704 bp, of which 68 variable sites and 79 haplotypes were identified (Table 2). Sympatric L. lamarum, containing 18 specific variable sites, possessed specific haplotype H20 and could be distinguished clearly from other taxa. Thirty-six species-specific variable sites were used to identify the occurrence of hybridization between L. duciformis and L. paradoxa. As listed in Table 2, 65.6% sequences of L. duciformis have the following haplotypes: H1, H2, H3, H4, H5, H10, H14, H60, H62. Likewise, 84.8% sequences of L. paradoxa have haplotypes: H27, H43, H50, H54, H55, H57. For L. ×maoniushanensis, 63.2% sequences possessed haplotypes from L. duciformis and L. paradoxa (Table 2).

Table 2 Distribution of ITS haplotypes and cpDNA haplotypes in two hybrid swarms.

Locality	Taxa	Individuals of direct sequencing	Nuclear Haplotypes (No. of direct and cloned sequencing)	Individuals of cloned sequencing for ITS	cpDNA Haplotypes (No. of individuals)	
Mt. Maoniu	L. duciformis	15	H1(3), H2(4), H5(1), H9(1), H35(1), H50(1), H58(1), H59(1), H60(5), H61(1), H62(2), H63(1), H64(1), H65(1)	6/15	H1(15)	
	L. ×maoniushanensis	9	H2(1), H17(2), H22(3), H23(1), H27(3), H29(1), H35(4), H41(1), H42(1), H48(1), H50(1), H60(2), H66(1), H67(1), H68(1), H69(1), H70(1), H71(1), H72(3), H73(1), H74(1), H75(1), H76(1), H77(1)	9/9	H4(9)	
	L. paradoxa	20	H22(1), H27(4), H43(2), H50(13), H55(1), H56(1), H78(1), H79(1)	4/20	H4(20)	
	L. lamarum	7	H20(7)	0	H6(7)	
Heihai Lake	L. duciformis	20	H1(12), H2(3), H3(3), H4(2), H5(4), H6(1), H7(1), H8(1), H9(1), H10(2), H11(1), H12(1), H13(1), H14(2), H15(1), H16(1), H17(1), H18(1), H19(1)	10/20	H1(19), H2(1)	
	L. ×maoniushanensis	10	H5(3), H13(1), H14(2), H21(1), H22(8), H23(2), H24(1), H25(1), H26(1), H27(1), H28(1), H29(1), H30(1), H31(1), H32(4), H33(1), H34(1), H35(2), H36(1), H37(2), H38(1), H39(1), H40(1), H41(1), H42(1), H43(1), H44(1), H45(1), H46(1), H47(1), H48(1), H49(1), H50(1), H51(1), H52(1), H53(1)	10/10	H1(6), H4(3), H5(1)	
	L. paradoxa	20	H27(5), H35(1), H50(7), H54(2), H55(4), H56(1), H57(2)	2/20	H4(20)	
	L. lamarum	6	H20(6)	0	H3(6)	

Haplotype networks for L. duciformis, L. paradoxa, L. ×maoniushanensis and L. lamarum were generated to illustrate their relationships (Fig. 2). The nrITS network contained three major subnetworks, where individuals of L. duciformis and L. paradoxa constituted two major subnetworks (here termed parts A and B), respectively, and sympatric L. lamarum individuals formed a third subnetwork (termed part C) located at the edge of the nrITS network (Fig. 2). Haplotypes derived from the cloned sequences of L. ×maoniushanensis were scattered among parts (A) and (B), which corresponded to L. duciformis and L. paradoxa, respectively, among which some haplotypes were unique to L. ×maoniushanensis. These unique haplotypes may have resulted from unsampled polymorphism in the parental species, mutation in hybrid individuals, and/or genetic recombination in the hybrids.

Figure 2 Haplotype network inferred from combined nrITS data.

Each rectangular area represents one nrITS haplotype, and a black circle represents an inferred absent haplotype. Red and green symbols represent sequences of (A) parental L. duciformis and (B) L. paradoxa. Black and blue symbols represent sequences of putative hybrids (A and B) and sympatric L. lamarum (C), respectively. (A) Population locality: M, Mt. Maoniu; H, Heihai Lake; (B) taxa: D, L. duciformis; P, L. paradoxa; M, L. ×maoniushanensis; L, L. lamarum. Numbers following taxon initials are sample numbers and clone numbers (if any).

Analyses of the combined cpDNA sequences

The aligned and trimmed length of the amplified psbA-trnH, trnL-rpl32 and trnQ-5′rps16 regions were 453 bp, 916 bp and 905 bp, respectively. The concatenated length for three fragments was 2,275 bp, containing 19 variable sites in total (Table S3). Among loci, trnL-rpl32 had the most variable sites (52.6%). Ligularia ×maoniushanensis shared 31.6% variable sites with L. duciformis and 63.2% with L. paradoxa. Six haplotypes were identified among all the individuals (H1-H6; Table 2 and Table S3). A haplotype network was inferred based on the three combined cpDNA fragments of L. duciformis, L. paradoxa, L. ×maoniushanensis and L. lamarum (Fig. 3). Ligularia duciformis had two haplotypes (H1 and H2) and L. paradoxa exclusively had the H4 haplotype. Individuals of L. ×maoniushanensis at Mt. Maoniu shared haplotype H4 with L. paradoxa. However, at the Heihai Lake locality, 60% of L. ×maoniushanensis individuals shared haplotypes H1 with L. duciformis, 30% shared haplotype H4 with L. paradoxa, and 10% (sample HM5) had a unique H5 haplotype. Haplotypes H3 and H6 were unique to L. lamarum; each population of L. lamarum had a single haplotype (Fig. 3; Table 2).

Figure 3 Haplotype network inferred from cpDNA data.

Each rectangle represents one haplotype, and the black circles represent haplotypes not detected. Red and green symbols represent sequences of parental L. duciformis (A) and L. paradoxa (B), respectively. Black and blue symbols represent sequences of putative hybrids (A and B) and sympatric L. lamarum (C), respectively. (A) Population locality: M, Mt. Maoniu; H, Heihai Lake; (B) taxa: D, L. duciformis; P, L. paradoxa; M, L. ×maoniushanensis; L, L. lamarum. Numbers following taxon initials are sample numbers (if any).

SSR profiles

Eleven SSR loci were genotyped for L. duciformis, L. ×maoniushanensis, and L. paradoxa, with 69, 53, and 72 alleles identified, respectively. The average observed heterozygosity was 0.624, 0.696 and 0.581 for L. duciformis, L. ×maoniushanensis, and L. paradoxa, respectively.

STRUCTURE analyses

In the STRUCTURE analysis, SSR datasets from the two locations were analysed jointly. The highest ΔK value was obtained with K = 2 for the putative parents and hybrids (Table S4; Fig. S1), indicating the presence of two genetic clusters. For L. duciformis, all individuals (15) from Mt. Maoniu and most individuals (18/20) from Heihai Lake were assigned to one purebred cluster (qi > 0.90; Fig. 4A). For L. paradoxa, all individuals (18) from Mt. Maoniu and most individuals (17/18) from Heihai Lake were assigned to the other purebred cluster (qi > 0.90). However, a few individuals that were morphologically identified as L. duciformis or L. paradoxa were assigned to the mixed cluster with high probability (qi (HD18) = 0.828, qi (HD20) = 0.885, qi (HP15) = 0.880). For L. ×maoniushanensis, eight individuals from Mt. Maoniu and nine from Heihai Lake were inferred to have a genetic composition deriving from both L. duciformis and L. paradoxa, and were unassigned to either cluster (0.10 < qi < 0.90). The remaining two individuals of L. ×maoniushanensis were assigned to the L. paradoxa cluster (qi > 0.90).

Figure 4 Model-based clustering analyses by STRUCTURE based on SSR markers with K = 2.

Samples from all individuals sampled from two locations (A), Mt. Maoniu (B) and Heihai Lake (C) were analysed, respectively. Vertical bars represent individuals and probabilities of assignment to each cluster.

The STRUCTURE analyses for SSR data treating each of the two sampling sites individually generated very similar results. STRUCTURE analyses of the three taxa from Mt. Maoniu indicated that the optimal number of clusters was K = 2 (Table S5 and Fig. S2). Fifteen individuals of L. duciformis and 18 individuals of L. paradoxa were assigned to the two purebred clusters (qi > 0.90), and nine individuals of L. ×maoniushanensis were assigned to the mixed cluster (0.10 < qi < 0.90) (Fig. 4B). Two genetic clusters (K = 2) was also suggested for taxa from Heihai Lake (Table S6; Fig. S3). Twenty individuals of L. duciformis and 17 individuals of L. paradoxa were allocated to the two purebred clusters (Fig. 4C). Most individuals of L. ×maoniushanensis (8/10) showed an estimated membership (qi) ranging from 0.16 to 0.84, and assigned to the admixed group, whereas the two remaining individuals were considered to be L. paradoxa.

In comparing the results of joint with separate analyses, we found some individuals with values near the critical value were assign to mixed class in joint analyses, e.g., qi (HD18) = 0.828, qi (HD20) = 0.885, however, they were assign to parental class (qi (HD18) = 0.924, qi (HD20) = 0.918, respectively) in separate analyses.

Figure 5 Posterior probability distribution of SSR data with NewHybrids.

All samples are represented by two vertical bars partitioned into segments whose lengths are proportional to the likelihood of belonging to a certain class. MD and HD, MM and HM, and MP and HP represent morphologically identified L. duciformis (LD), L. ×maoniushanensis and L. paradoxa (LP), respectively. Samples from Mt. Maoniu (A) and Heihai Lake (B) were analysed, respectively. M and H represent the two hybrid swarms, Mt. Maoniu and Heihai Lake, respectively.

NewHybrids analyses

NewHybrids analyses were conducted for 89 individuals from the two sampling areas; the joint analyses (Fig. S4) were slightly different from the separate analyse for each sampling area (Fig. 5). For the Mt. Maoniu population, with the exception of one individual (MD15), all the individuals of L. duciformis and L. paradoxa were assigned to one parent with high PP (>0.94; Fig. 5A). Individual MD15 of L. duciformis may be a later hybrid progeny (PP < 0.90). Individuals of L. ×maoniushanensis were F2 hybrids with relatively high PP (>0.80). For the Heihai Lake population, 18 out of 20 individuals of L. duciformis and 13 out of 17 individuals of L. paradoxa were assigned to their respective parental class with high PP (>0.90; Fig. 5B). The other two individuals of L. duciformis were not assigned to L. duciformis (PP < 0.90). Furthermore, the remaining four individuals of L. paradoxa were not assigned to any class (PP < 0.50), and these individuals might be backcrosses to L. paradoxa. For L. ×maoniushanensis, eight individuals were F2 hybrids with a posterior probability higher than 0.50, whereas the other two individuals were not assigned to any class (PP < 0.50) which might result from backcrossing to L. paradoxa.

INTROGRESSION analyses

Figures of genomic clines (Figs. 6A–6K) were generated based on SSR data. This method unmasked marked heterogeneity in locus-specific patterns of introgression, and genomic clines of introgressed alleles can be identified. Eight loci (L38, L40, L41, L64, S4, S23, S24 and V40) showed significant deviations after FDR correction from expectation value based on a null model of neutral introgression (P < 0.021), indicating that introgression was detected in these loci (Fig. 6). For the six loci (L40, L64, S4, S23, S24, and V40), the homozygote and heterozygote genomic clines were steeper than predicted by the neutral model. Genotypic variation at locus L64 was also involved in steep transitions between parental homozygotes, but very few heterozygotes were observed.

Figure 6 Genomic clines for nuclear 11 SSR loci (A–K) from hybrids between L. duciformis and L. paradoxa.

The name of each locus is given, as is the P value (P < 0.021 indicates significance after FDR correction) for the test of departure from neutral expectations, on each panel. Solid and dashed clines represent the 95% confidence intervals for the expected homozygotes (Ad/Ad or Ap/Ap; dark green) and heterozygotes (Ad/Ap; light green) genomic clines given neutral introgression. The solid and dashed lines give the estimated clines based on the observed homozygotes and heterozygotes, respectively. Circles indicate the raw genotypic data (L. duciformis homozygotes (Ad/Ad): red, heterozygotes (Ad/Ap): yellow, and L. duciformis homozygotes (Ap/Ap): black), with counts of each on the vertical axis. The hybrid index quantifies the fraction of alleles derived from L. paradoxa across all 11 markers.

There was some heterogeneity among loci in the patterns of introgression between species, as well as some similarities, which was visually evident (Fig. 7A). On each locus, permutations of three genotypes, Ad/Ad, Ad/Ap and Ap/Ap, were observed. Genotypes with more Ad/Ap were mainly confined to the loci L40, L77, S2, S4, S23, S24, Sm083 and V40. Some individuals of L. duciformis shared genetic composition with L. paradoxa and L. ×maoniushanensis at most loci, except for L40, L64 and S24, and L. paradoxa shared genetic composition with most loci, except for L41. Moreover, genomic cline analysis revealed a gradually increasing pattern of introgression among L. ×maoniushanensis individuals in the proportion of L. paradoxa ancestry estimated (Fig. 7B).

Figure 7 Overview plot of patterns of introgression for all markers and individuals in an admixed population.

Markers are ordered based on map locations (A). Each rectangle corresponds to an individual’s genotype at a given locus: dark green, indicating L. duciformis homozygotes (Ad/Ad), to light green indicating L. paradoxa homozygotes (Ap/Ap); heterozygote (Ad/Ap) genotypes are represented by intermediate green blocks. White blocks indicate missing data. (B) is a plot of the fraction the genome inherited from L. paradoxa ancestry.

ISSR profiles

Nine polymorphic ISSR markers were screened for the sampled individuals of L. duciformis, L. ×maoniushanensis, and L. paradoxa. Among these individuals, L. duciformis had 12 unique polymorphic loci and L. paradoxa had nine, whereas L. ×maoniushanensis did not contain any unique loci. Five loci that were shared among three Ligularia taxa were excluded.

NewHybrids analyses

NewHybrids analyses were separately implemented for each sampling site. For individuals from Mt. Maoniu, with the exception of one individual (MD3), all the individuals of the putative parents were assigned to L. duciformis and L. paradoxa with high PP values (>0.939). Moreover, all the putative hybrids (L. ×maoniushanensis) exhibited the highest PP (>0.749) for the F2 hybrids compared to other genotype classes. For the Heihai Lake population, most individuals of L. duciformis (18/20) and L. paradoxa (12/17) were assigned to genotypes consistent with morphological identification (Fig. 8B). Two individuals of L. duciformis and one individual of L. paradoxa were not assigned to a parental class; this was with low PP (<0.90). The remaining four individuals of L. paradoxa were not assigned to any class (PP < 0.50). For L. ×maoniushanensis, nine individuals were assigned to F2 hybrids with the PP > 0.50, and the remaining one was assigned to backcross to L. paradoxa.

Figure 8 Posterior probability distribution of ISSR data using NewHybrids.

All the samples are represented by two vertical bars partitioned into segments whose length are proportional to the likelihood of belonging to a certain class. MD and HD, MM and HM, and MP and HP represent morphologically identified L. duciformis (LD), L. ×maoniushanensis and L. paradoxa (LP), respectively. Samples from Mt. Maoniu (A) and Heihai Lake (B) were analysed, respectively. M and H represent the two hybrid swarms, Mt. Maoniu and Heihai Lake, respectively.

Discussion

Evidence for natural hybridization and introgression: nrITS and cpDNA markers

In this study, the nrITS data support the hypothesis that morphologically intermediate individuals are primarily hybrid descendants between L. duciformis and L. paradoxa and that the sympatric taxon L. lamrum does not participate in hybridization. In the nrITS analyse, some morphologically identified parental individuals showed double peaks of nrITS; the cloned nrITS sequences of these individuals suggest that these individuals may be the progenies of backcrossing. Backcrossing has also been suggested previously to cause frequent gene flow between Ligularia species, giving rise to complex relationships between them (Yu, Kuroda & Gong, 2011). In our results, some individuals had unique haplotypes; such variants might be caused by hybridization between parents containing different ITS sequences (Baldwin et al., 1995; Sang, Crawford & Stuessy, 1995; Li et al., 2014) or independent evolution of haplotypes through mutation exceeding the rate of concerted evolution or pseudogenization among ITS copies (Feliner & Rosselló, 2007; Kosnar et al., 2012).

Since cpDNA is reported to be maternally inherited in Ligularia (Zhang & Liu, 2003), analyses of chloroplast sequences can reveal the direction of hybridization. We observed that L. paradoxa was the only maternal species in Mt. Maoniu, whereas L. duciformis and L. paradoxa were both maternal species in Heihai Lake. Consequently, we deduce that the hybridization was primarily unidirectional (L. duciformis ♂ × L. paradoxa ♀) in Mt. Maoniu and bidirectional in Heihai Lake. Moreover, the hybrid combination of L. duciformis ♀ × L. paradoxa ♂ was predominant at Heihai Lake. These results congruent with those of Pan et al. (2008) in Mt. Maoniu where natural hybridization between L. duciformis and L. paradoxa was exclusively unidirectional; however, the results also suggest that hybridization in Heihai Lake is asymmetrical, with L. duciformis as the primary maternal parent.

Given potentially influential factors such as habitat, population size, and gametophytic-sporophytic interactions during fertilization or organelle-nuclear gene interactions, asymmetric hybridization barriers in plants are common (Tiffin, Olson & Moyle, 2001; Turelli & Moyle, 2007). The phenomenon of asymmetrical hybridization has been reported in many taxa, such as Rhododendron (Ma et al., 2010; Yan, Gao & Li, 2013; Zha, Milne & Sun, 2009) and Melastoma (Liu et al., 2014). According to our field observation, individuals of L. duciformis are more abundantly distributed than any other sympatric Ligularia species in both hybrid swarms, possibly swamping available conspecific individuals. In addition, Mt. Maoniu and Heihai Lake have different habitat conditions. Mount Maoniu, possessing abundant wet soil under moist forest, appears to be more mesic habitat than Heihai Lake, which has less water availability among arid rock crevices where the plants occur. In the more mesic habitats of Mt. Maoniu, L. duciformis serves as paternal species, but in the arid habitats at Heihai Lake, L. duciformis and L. paradoxa are both maternal species. These different population dynamics may result from habitat or climatic factors, yet little is known of habitat requirements in these species and further studies are needed to clarify the most important factors in the present case.

Evidence for natural hybridization and introgression: SSR and ISSR markers

Given the evidence for substantial interspecific gene flow accrued in this study, it could be alternatively hypothesized that these Ligularia species distributed at the two study sites comprise a single large, randomly breeding complex to be treated as a single genetically distinct species. This hypothesis can be directly tested by the STRUCTURE analyses of the SSR data of all individuals from the two locations. The random-breeding complex hypothesis would be supported if L. duciformis and L. paradoxa cluster by sampling sites rather than by species, since gene flow is expected among individuals in sympatry. STRUCTURE analyses indicated that individuals from the two sampling sites cluster by species rather than by sampling sites, indicating the presence of genetic differentiation among these species; hence considering the parents as separate species is justified given the loci examined here.

Although overall results were congruent among datasets and analyses, we found that a few individuals were assigned to different genotype categories for purebred and hybrids in STRUCTURE and NewHybrids. This can be explained as a result of marker choice; accurate identification of genetic categories using molecular methods depends on the markers selected and the degree of differentiation among species (Anderson & Thompson, 2002; Vähä & Primmer, 2006). We employed a default threshold (qi, 0.90) for discriminating purebred and hybrid specimens in our study, which in combination with differing evolutionary patterns among markers may lead to discrepancies among individual data partitions.

We also tested the hybrid origin of L. ×maoniushanensis and its genetic structure. In STRUCTURE analyses, L. duciformis and L. paradoxa formed two distinct genetic clusters; many individuals of L. ×maoniushanensis showed genetic admixture between the two clusters, implying some F1 hybrids. Based on the NewHybrids analyses of the SSR markers, most individuals of L. ×maoniushanensis were assigned to as F2 hybrids or backcrosses. The thresholds used for assigning individuals to the different genetic categories are different between STRUCTURE and NewHybrids. It is possible that different hybrid categories were suggested by the two methods partly for this reason. In addition, these two methods have different strengths. STRUCTURE is more efficient for evaluating the presence of hybrids in wild populations (Marie, Bernatchez & Garant, 2011). However, given the assumption of two parental categories, NewHybrids assigns explicit hybrid categories, which might be likely to show higher assignment accuracy than those obtained using STRUCTURE (Marie, Bernatchez & Garant, 2011). The NewHybrids results indicate that F2 hybrids and backcrosses are more prevalent than F1 hybrids. The occurrence of F2 hybrids and the preponderance of backcrosses to both parental species are consistent with recent hybridization and potentially ongoing hybridization between the two species and among hybrid derivatives. Although SSR and ISSR markers had different resolution for distinguishing among the genotypes of the examined samples, a similar result was obtained when SSR results were compared with the ISSR results in NewHybrids analyses. Therefore, these results provide compelling evidence for the hybrid origin of L. ×maoniushanensis.

Meanwhile, based on the STRUCTURE results, a few parental individuals of L. duciformis and L. paradoxa were genetically assigned to L. ×maoniushanensis, including HD18 and HP15, and a few individuals of L. ×maoniushanensis were genetically assigned as L. paradoxa, such as MM8 (from Mt. Moniu) and HM9 (from Heihai Lake). These individuals, for which morphology and molecular markers were incongruent, may result from frequent backcrosses to L. paradoxa, or the insufficiency of SSR markers to identify the individuals. In the NewHybrids analyses, some parental individuals with PP <0.90 are morphologically identified as L. duciformis and L. paradoxa, but they may be later generation hybrids according to the results of NewHybrids. Some hybrids that morphologically resemble parental species may result from continual backcrosses to the parental individuals. For example, one L. duciformis individual and one L. paradoxa individual in Heihai Lake were assigned to F2 or backcross hybrids; this indicates the existence of bidirectional introgressive hybridization between L. duciformis and L. paradoxa.

Results from INTROGRESS further confirmed the presence of extensive introgressive hybridization between L. duciformis and L. paradoxa. Genomic cline analyses revealed a diversity of introgression patterns among loci. Clines for the majority of markers were inconsistent with neutral introgression in hybrids (8/11; Fig. 6). Excess and deficits of the three genotypes (Fig. 7) presented different patterns among loci. This may be consistent with the action of selection at these loci; the potential presence of non-neutral introgression in these populations requires further research.

From infertile seeds to repetitive backcrosses or hybrids

Seed germination rates in L. ×maoniushanensis have been examined previously; L. ×maoniushanensis was found to completely lack viable seeds (Pan et al., 2008). However, this experiment does not completely exclude the existence of fertile seeds from hybrids. It is possible that rare fertile seeds are occasionally generated from repeated hybridization events. The lower fertility of the initial hybrids compared to parental individuals could imply that hybrid progenies play a minor role in the evolution of a given species complex (Arnold, 1997). Although the fertility of the hybrids may limit the production of hybrid individuals, as initial hybrids repeatedly form, the opportunity of producing later hybrid generations increases, potentially restoring fertility in late-generational hybrids. A classic case comes from a repetitive cross experiment between Helianthus annuus and H. petiolaris (Asteraceae) which resulted in an increase of fertile seed (Heiser et al., 1969). Although low levels of fertility and viability often occur in initial hybrid generations, the fertility for late-generational hybrids could result in a stabilized hybrid species (Rieseberg, 1991). Therefore, the production of some fertile hybrid progenies is possible during hybridization between L. duciformis and L. paradoxa given the frequent occurrence of hybrid individuals.

Is L. ×maoniushanensis a hybrid species or a hybrid swarm?

Given the individuals we sampled, the status of L. ×maoniushanensis is unclear. A true hybrid species (an independent lineage of hybrid origin) should show patterns of continuous breeding among its populations at least beyond the F2 generation (Zhou, Shi & Wu, 2005). Based on our observations, the present number of L. ×maoniushanensis individuals in the Mt. Maoniu is smaller than a decade ago. This phenomenon could be explained by the low fertility of hybrids and a preponderance of early-generational hybrids. Frequent hybridization and repetitive formation of L. ×maoniushanensis in these locations does not appear to have significantly increase the population size. However, L. ×maoniushanensis maintains a sympatric distribution with the parental species rather than occupying a distinct habitat or niche, and frequent backcrossing may inhibit the formation of reproductive isolation with parental species. The evolutionary species concept (Simpson, 1951; Blackwelder, 1962; Soltis & Soltis, 2009; Wheeler, 2000; Wiley, 1978) delimits as species those lineages evolve separately and have their own evolutionary tendencies. Under this concept, we conclude that L. ×maoniushanensis is repeatedly generated from the parental species, and since it is not independently evolving from its parental taxa it should not be considered as a new species. However, the intrinsic factors driving the directionality and asymmetrical contributions of parental genomic material, a more precise determination of hybrid generations, and the affinity of assigned individuals should be the topics of further investigation.

Conclusions

Our study confirmed the hybridization between L. duciformis and L. paradoxa in two different locations by analysing sequences of nrITS and three chloroplast DNA regions. The SSR and ISSR markers also demonstrated extensive introgressive hybridization and provided convincing evidence for the origin of these putative hybrids between L. duciformis and L. paradoxa. Hybridization was inferred to be primarily unidirectional (L. duciformis ♂ × L. paradoxa ♀) in Mt. Maoniu and bidirectional in Heihai Lake, which was predominant by L. duciformis ♀ × L. paradoxa ♂. Due to frequent hybridization and introgression in these regions, these hybrids do not appear to be reproductively isolated from parental species or to represent independent lineages. Accordingly, we identify L. ×maoniushanensis individuals as members of a hybrid swarm, potentially representing the introgression of traits from one species to another, rather than a hybrid speciation event.

Supplemental Information

Table S1 Details of taxon’s sample locations and sample size (n) of two regions

Leaf samples and voucher specimens were collected from two naturally occurring admixed-growing taxa of L. duciformis, L. paradoxa and their putative hybrids in two hybrid zones (Mt. Maoniu and Heihai Lake, Ninglang County, Yunnan of China) between August 2013 and June 2014. These putative hybrids were morphological intermediates and variant individuals, and the degrees of cracking palmatisect of the leaf blades were different, ranging from lobed to deeply lobed. The suspected backcrossing individuals were found and their morphologies were close to one of the parents. The sympatric species, L. lamarum, was also collected from these two locations to confirm whether it participated in the hybridization.

Click here for additional data file.

Table S2 The number of clones sequenced for each accession

Click here for additional data file.

Table S3 The sites of variation and indels of three cpDNA sequences in related materials

Click here for additional data file.

Table S4 Analysis of appropriate K value for the SSR data of three Ligularia taxa on the Mt. Maoniu and Heihai Lake sampling sites

Click here for additional data file.

Table S5 Analysis of appropriate K value for the SSR data of three Ligularia taxa on the Mt. Maoniu sampling site

Click here for additional data file.

Table S6 Analysis of appropriate K value for the SSR data of three Ligularia taxa on the Heihai Lake sampling site

Click here for additional data file.

Figure S1 The relationship of SSR data between K values and △K for the combined Mt. Maoniu and Heihai Lake data sets

Click here for additional data file.

Figure S2 The relationship of SSR data between K values and △K for the Mt. Maoniu data set

Click here for additional data file.

Figure S3 The relationship of SSR data between K values and △K for the Heihai Lake data set

Click here for additional data file.

Figure S4 Posterior probability distribution of SSR data by using the NewHybrids program

NewHybrids analysis was implemented for 89 individuals from the two sampling sites showing slightly different results of join. All the samples are represented as a vertical bar partitioned into segments whose length is proportional to the likelihood of belonging to a certain class. MD and HD, MM and HM, MP and HP represent morphologically identified L. duciformis (LD), L. ×maoniushanensis and L. paradoxa (LP), respectively. M and H represent the two hybrid zones, Mt. Maoniu and Heihai Lake, respectively.

Click here for additional data file.

Supplemental Information 1 All ITS sequences

Click here for additional data file.

Supplemental Information 2 All psbA-trnH sequences

Click here for additional data file.

Supplemental Information 3 All trnL-rpl32 sequences

Click here for additional data file.

Supplemental Information 4 All trnQ-5’rps16 sequences

Click here for additional data file.

Supplemental Information 5 All sequences-IDs from GenBank

Click here for additional data file.

We thank Ying Zheng, Ningning Zhang, Jian Liu, Jiaojun Yu, Hafiz Muhammad Wariss, and Moses Cheloti Wambulwa from Kunming Institute of Botany (KIB), Chinese Academy of Sciences, for help with revising the manuscript in English. We appreciate Yuezhi Pan and Qitai Zhang from KIB for help with samples collection. We are also grateful to anonymous reviewers for detailed and constructive comments.

Additional Information and Declarations

Competing Interests

Author Contributions

DNA Deposition

Data Availability

The authors declare there are no competing interests.

Rong Zhang performed the experiments, analyzed the data, contributed reagents/materials/analysis tools, wrote the paper, prepared figures and/or tables, reviewed drafts of the paper.

Xun Gong conceived and designed the experiments, contributed reagents/materials/analysis tools, reviewed drafts of the paper, obtain fundation to pay the fee of sequence and materials.

Ryan Folk reviewed drafts of the paper.

The following information was supplied regarding the deposition of DNA sequences:

All sequences are deposited via GenBank (accession numbers KY307306–KY307785).

The following information was supplied regarding data availability:

The raw data has been supplied as a Supplementary File.

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
