# Peer review of "Evidence for continual hybridization rather than hybrid speciation between Ligularia duciformis and L. paradoxa (Asteraceae)"

_PeerJ, doi:10.7717/peerj.3884_

## Round 0.1 · original submission · Major Revisions

Please pay special attention on correcting the grammar and style of writing, as both reviews noted that it tends to prevent understanding of the manuscript. Also, play special attention to the interpretation and construction of the phylogenetic tree in the manuscript. Please heed all the reviewers' comments.

Reviewer 1 ·

Basic reporting

The grammar needs attention (e.g., the first sentence is a sentence fragment, and the word “still” makes the title of the paper confusing—it suggests that something about the nature of Ligularia x maon… has changed). In some places the grammatical problems make this manuscript difficult to follow, and thus difficult to review. Some of the ideas present may benefit from clarification/substantiation, but that’s hard to assess currently.

Considering hybridization “one of the drivers of speciation” is a little bit strong for the first sentence of the paper—that’s a somewhat controversial/complex claim.

Ln78 – provide citation for these species belonging to this section. And introduce the family affinities of this genus at some point?

The introduction might be more effective if it were more tightly organized/clarified. E.g., a section on the evolutionary significance of hybridization, followed by a section on Ligularia, followed by a section on the specific study system, and its relation to the hybridization question.

Ln94 – it’s unclear at this point what the distinction is between “natural hybrid species” and “hybrid offspring.”

The voucher information in table S1 is incomplete. Each studied population should be supported by at least one voucher, deposited in an official herbarium (collector number and herbarium of deposition need to both be indicated here).

The beginning of the Discussion belongs in the Introduction.

Fig. 1 – it would be more informative to present a most-parsimonious tree (rather than a consensus tree) and to then label the branches of that tree with the bootstrap support values. Strict consensus trees don’t have very much information in them.

Fig. 2 – Interesting (odd?) that relatively few of the hybrid sequences are the same as the most-common haplotype from either parent. Instead, many of the hybrid sequences are unique. Is this an artifact of the fact that the sequences from the hybrids were cloned (and thus have more PCR errors, etc., in them), and the parental accessions were direct-sequenced?

Fig. 3 – Something is wrong with this tree. Is it also a consensus tree? Instead, a single most-parsimonious tree should be presented, with meaningful branchlengths, and bootstrap support values added on.

Fig. 5 – This figure would be clearer if each of the parental species got the same color in each of the panels.

A figure showing some plants of each taxon would be a welcome addition to the paper.

Experimental design

It’s not clear (at least not in the beginning) how the cloned data were used. How many colonies were sequenced? For individuals with two “alleles”, how were they analyzed?

ln141 – 1000 replicates for the best tree search? (Rather than for bootstrapping).

Validity of the findings

224 – “almost all individuals” of the hybrid shared haplotypes with one of the putative parents. But not all? What does that imply?

Reviewer 2 ·

Basic reporting

The English language should be improved to ensure that your international audience can clearly understand your text. I suggest that you have a native English speaking colleague review your manuscript. Some examples where the language could be improved include lines 62-65, 75-75 among others – the current phrasing makes comprehension difficult.
Line 103 needs more precision in concepts. “Classification” is part of a different taxonomic task. It should say: “identified” or “determined”.
Some grammar errors like spaces, periods, etc. are marked in the text.
The rest of the information about literature, figures, data, etc. have no problem.

Experimental design

Minor comments and suggestions: The plant materials acquisition must be more explicit. (lines 98-102, and 179). Say how did the authors got the F1, and F2 materials. Do specimens on F1 and F2, and from backrosses presented morphological variation?

I suggest to include the percentage of the species individuals, instead of the word "most" (line 124).

Validity of the findings

The authors are not correctly interpreting the phylograms. The explanation or interpretation in lines 228-230 of the Fig. 1 are not correct. What the authors consider as "cluster A", it is an unresolved clade formed by three different groups, and a polytomy with 22 unresolved individuals. The nrITS outcome and the provided interpretations based on the tree are erroneous.
On the other side, the description of Fig. 3 in lines 253-257 is also incorrect. The tree does not show independent clusters for L. spicata (outgroup), L. lamarum, and L. paradoxa x maoniushanensis. Even more, what they consider a clade for L. lamarum (H3 and H6) it is part of the unresolved area of the tree. You cannot consider that H3 and H6 show a closer relationship than the rest of individuals sampled.
The evidence and discussion for nrITS and cpDNA evidence for the Ligularia species hybridization and introgression patterns and processes must be reinterpreted (Some information about this can be found easyli in http://evolution.berkeley.edu/evolibrary/article/phylogenetics_03,
in http://tolweb.org/tree/home.pages/treeinterpret.html, or in Lanford, M., et al. 2014. Bioinformatics 30 (17): i519-i526. DOI:
https://doi.org/10.1093/bioinformatics/btu463

The rest of the evidence seems to support their concluisons.

Additional comments

Tha manuscript present a very interesting study to understand plant hybridization with molecular markers and analysis in an evolutionary context. However part of the phylogenetic interpretations are not correct. If they are willing to reinterpret the nrITS and cpDNA phylograms, the results, discussion and conclusions will be substantially improved.

Annotated reviews are not available for download in order to protect the identity of reviewers who chose to remain anonymous.

---

## Round 0.2 · Minor Revisions

Please heed the remaining comments from the reviewers when preparing the revised manuscript.

Reviewer 1 ·

Basic reporting

Grammar and English usage need further attention. References, figures, data sharing, results/hypotheses -- all seem fine (I suggest removing at least one figure -- see more detailed comments below).

Experimental design

yes

Validity of the findings

yes

Additional comments

This paper is greatly improved over the earlier version, but the grammar still needs considerable attention (at least in my opinion – ultimately this would be up to the journal). For example, in the abstract, it should be “is a hybrid” instead of “is hybrids” in the results section, and it’s unclear what “sufficient for hybrid generation determination” means. In some areas of the paper the language issues are sufficient to make it very difficult to understand what is being said.

ln69 I still feel this section is too strong, even with the stated caveat that “not all hybrids form new species.” At the least, I’d suggest replacing “they will” with “they may” on this line

88 What is “family affinity”?

89 “affinities of many species at some point” is exceedingly vague

98 I don’t understand: “An interesting phenomenon in both populations is that some individuals are different from other, and they are typical individuals of the species.”

113 State this last point more clearly; something like: “Is L. xmao a cohesive hybrid species, reproductively isolated from its progenitors, or do the L. xmao individuals instead constitute a recurrently-formed hybrid swarm”

152 The number of clones sequenced for each accession should be included in a table if they haven’t been already.

159 You mean that a consensus sequence was used for closely-related sequences from a single individual? What cut-off was used to decide if sequences were closely enough related to merge?

163 All ITS cloned sequences were used? But then it says later that two sequences were selected. That selection process sounds suspicious – if the authors are deliberately selecting sequences that are close to the parental sequences and throwing out the rest, aren’t they biases the analyses towards the desired result?

265 Inclusion of identical sequences should not reduce resolution or support.

353 I’m not familiar with the introgression program but very small p-values typically indicate highly significant results, which I would have thought in this case would mean strong indication of introgression. That’s not the case?

383 The beginning of the discussion has already been covered in the introduction. The entire first section (“Conditions of…”) could be omitted.

469 Was the germination trail (1000 seeds) part of this study, or the earlier cited one?

475 etc But this mechanism – that largely infertile F1s are formed at high enough rates that a lucky few produce enough seed to allow hybrids to perpetuate – is inconsistent with your data, which shows almost no F1s but a lot of F2s.

508 This section on Pinus etc belongs in the introduction rather than in the discussion

Fig. 1 Nice!

Fig. 2 This figure is still strange. Why are all the terminal branches exactly the same length? Contrary to the caption, I very much doubt that this is a most-parsimonious tree. (At least, not a phylogram, with the associated scale bar [10 changes]). A single most-parsimonious tree should be presented, as a phylogram. For individuals that had multiple sequence types present, some method should be used to show which sequences come from which accession (e.g., a series of lines connecting sequences from a given individual).

Fig. 3 I don’t think this figure adds anything to what is already shown in Fig. 2. It is not a depiction of “reticulate history” unless I misunderstand. Instead it is a haplotype network, where reticulations indicate other possible mutation paths.

Fig. 6 Strange and interesting that there’s very little evidence of F1 plants in this sample (whereas F2s seem much more common). I don’t remember that message being made very clearly in the manuscript, but I might have missed it.

Fig. 7 I don’t know enough about this approach to be able to follow this figure. What are fAmAm and fAmAd?

Table 3 I think this table could safely be omitted.

Reviewer 2 ·

Basic reporting

Just typographic errors:
Line 43: say ecidence; must say evidence
Line 235: say maoniushan- ensis; must say maoniushanensis

Experimental design

no comments

Validity of the findings

Lines 227-230, and 235-141:
The tree presented in the Fig 1 is not resolved. When a phylogeny has no resolution, you cannot describe group relationships. In this case, it is not true that the outcome resolved two phylogenetic sections (A and B, even if they resulted in the network at Fig 2, which is not a phylogeny). Hence, what the authors call section B, has the same probability to be phylogenetically related with accession H2, or with the H5-H7 branch or with any other. The order of the branches resulting from the analysis does not gives hierarchical groupings or relationship evidence. If the clade branch is not resolved at the base, you cannot say that you have a monophyletic group. There are not sections to be separated, you can say that most of the times, some samples resulted together forming branches (indicate bootstrap numbers or parsimony branches support), but that you do not know the relationships among them. I cannot agree with the tree description, interpretation, and conclusions presented.

Additional comments

Most of the evidence presented in the paper support the proposal of hybridization and introgression within Ligularia duciformis and L. paradoxa. Then I suggest to leave the phylogenetic analysis out of the paper since is not a good evidence to support the hypothesis presented in this study.

---

## Round 0.3 · Minor Revisions

Please have the manuscript revised for English and grammar, so that once the English is deemed acceptable, it can be accepted in PeerJ.

Reviewer 1 ·

Basic reporting

I realize that this isn't particularly helpful, but the English grammar, etc., is still not at a state that I would be comfortable with, were I a journal editor. Perhaps peerJ can do some copy editing?

Experimental design

no comment

Validity of the findings

no comment

Additional comments

no comment

Reviewer 2 ·

Basic reporting

no comment

Experimental design

no comment

Validity of the findings

The problems I found are solved. I think that the paper migth be published as the last version is.

Additional comments

No comment

---

## Round 0.4 · Minor Revisions

The revised manuscript is improved from previous versions; however, the English is still not up to par. Please revise and improve the English in the manuscript, as it still harbors many mistakes which hamper reading and understanding.

---

## Round 0.5 · Minor Revisions

Please improve the English substantially, as this is now all that is needed for the manuscript to be accepted in PeerJ. If you do not improve the English (a very good option is to seek professional help on the matter from one of the many companies that perform this service), the manuscript may have to be rejected.

---

## Round 0.6 · accepted · Accept

Please do a final overview of the text, as for example, in line 139 is "are available in Genbank" and not "were available in Genbank".